# EXPLORING INSTRUCTION DATA QUALITY FOR EXPLAINABLE IMAGE QUALITY ASSESSMENT

## ABSTRACT

In recent years, with the rapid development of powerful multimodal large language models (MLLMs), explainable image quality assessment (IQA) has gradually become popular, aiming at providing quality-related descriptions and answers of images. To achieve this goal, recent methods seek to construct a large-scale instruction tuning dataset to empower the MLLM with quality perception ability following the well-known scaling law. However, a large amount of instruction tuning data may cause substantial computational costs and redundant data, which in turn will cause harm to the performance of the model. To cope with this problem, in this paper, we challenge the scaling law and systematically investigate the role of data quality of the instruction tuning dataset for explainable IQA. Using a powerful pre-trained MLLM, we first investigate the changes in model performance after fine-tuning with different sizes of instruction tuning data. We find that selecting a subset of the data set randomly using an appropriate ratio can even lead to better results than training with the entire instruction tuning dataset, demonstrating the redundancy of current explainable IQA instruction tuning data. Beyond randomly sampling a subset, we propose a clustering-based data selection framework with three stages: clustering feature extraction, cluster quota allocation, and cluster sampling strategy. Then we systematically analyze the choices of each stage and propose a simple but efficient data selection method IQA-Select for explainable IQA. The experimental results demonstrate that IQA-Select can achieve 102.1% and 103.7% performance of full fine-tuning using only 10% selected data in Q-Bench and AesBench respectively, significantly reducing computational costs while achieving better performance. We hope that our paper can provide a new perspective for future research on exploring the quality of instruction tuning data for explainable IQA.

## 1 INTRODUCTION

In recent years, multimodal large language models have demonstrated powerful and generalizable visual understanding capabilities, and they have been widely applied to a broad range of computer vision tasks. In light of these facts, the visual quality assessment community is adapting MLLMs with quality-related instruction tuning data for explainable image quality assessment (explainable IQA).

To cope with the problem, current researchers mainly follow the scaling law of data and seek to a large-scale instruction tuning dataset with hundreds of thousands of samples. For example, the Q-Instruct (Wu et al., 2024a) dataset constructs about 200K examples with huge quality-related question-answering pairs, significantly boosting the performance of the MLLM model in visual quality perception task. The Aesexpert (Huang et al., 2024) dataset contains 409K multi-typed instructions to enable MLLM with better aesthetic capabilities. However, directly fine-tuning the MllM model with these large-scale dataset will introduce substantial computational costs and overfitting. Moreover, with the rapid development of the current MLLMs, the basic ability of the model is becoming more and more powerful, most of the samples in instruction tuning dataset may become a piece of cake for MLLM to learn. Hence it is a remaining problem that do we still need so much dataset for MLLM fine-tunine?

In light of these facts, we explore two meaningful questions and their answers in this paper: Firstly, **Do we really need massive instruction tuning data samples for explainable IQA?** The answer is no. Utilizing comprehensive experiments, we discover that the current state-of-the-art Multimodal Large Language Model, *i.e.* InternVL3-Instruct can already achieve considerable performance on visual quality answering benchmark Q-Bench (Wu et al., 2023). Then we gradually reduce the scale of the instruction tuning dataset and find that utilizing only $5\%$ percentage of the Q-instruct dataset can achieve performance comparable to full-scale fine-tuning. We also observe that as the ratio of randomly selected data increases, the performance curve of the fine-tuned MLLM is approximately an inverted U-shaped curve, demonstrating that the scale of the instruction tuning data should neither be too large nor too small. The reason is because the current instruction-tuning data set for explainable IQA may contain low-quality or redundant examples. Fine-tuning the MLLM with a large redundant dataset will also cause the MLLM to overfit on them with abundant meaningless training data, while fine-tuning the MLLM with a too small dataset will cause the MLLM to learn nothing. Hence, a compact and informative coreset is suitable for MLLM fine-tuning.

Based on the first question and its observation, we come up with the second question, **How can we effectively select useful instruction tuning data?** Under the setting of selecting $10\%$ instruction tuning data, we explore the clustering-based data selection pipeline with three stages named IQA-Select to select the diverse and meaningful instruction tuning samples from the full dataset. In practice, the three stages include the clustering feature extraction, cluster quota allocation, and cluster sampling strategy. We comprehensively explore the possible strategies of each stage in our framework. For cluster features, we explore the effectiveness of 9 different features from both model-related features and model-independent features. For cluster quota allocation, we explore the effectiveness of 11 allocation strategies derived from cluster density, cluster transferability, and instruction relevance score. For cluster sampling, we explore the effectiveness of using greedy mmd sampling, SVD sampling and PCA sampling. After comprehensive experiments, our final IQA-Select method utilizes the combination of MLLM features and vision text features for clustering, the combination of cluster transferability and density for quota allocation, and the SVD sampling strategy for cluster sampling. Our framework ensures both diversity and informativeness in the selected data.

Our proposed IQA-Select achieves excellent performance on explainable image quality assessment task and explainable image aesthetic assessment task. With 10 % selected instruction tuning data, IQA-Select can achieve 102.1% and 103.7% performance of full fine-tuning using only $10\%$ selected data in Q-Bench and AesBench respectively, demonstrating the great potential of selecting meaningful coreset data in the explainable image quality assessment area.

In summary, the main contributions of this paper are:

- We provide the first systematic study of instruction data quality for explainable image quality assessment.

- We introduce a clustering-based selection framework IQA-Select with three stages: clustering feature extraction, cluster quota allocation, and cluster sampling, which can select meaningful data from the whole dataset.

- We achieve a new state-of-the-art performance in Q-Bench and AesBench with only $10\%$ selected instruction tuning data.

- We believe this work opens a new research direction for data-centric explainable IQA, where the focus shifts from constructing large instruction datasets to curating high-value and diverse data.

## 2 RELATED WORK

### 2.1 IMAGE QUALITY ASSESSMENT

Image Quality Assessment (IQA) is a long-standing problem, which aims to objectively evaluate the perceptual quality of images in a way that aligns with human visual perception. In recent years, IQA has achieved remarkable progress and become increasingly popular, driven by the emergence of numerous methods and datasets.

IQA methods can be broadly categorized into traditional score-based IQA methods and recent explainable IQA methods. Traditional score-based IQA methods focus on predicting a scalar quality score consistent with human subjective ratings, and are commonly classified into full-reference (FR) (Wang et al., 2004; Sheikh & Bovik, 2006; Zhang et al., 2011; 2018; Ding et al., 2020), reduced-reference (RR) (Wang & Simoncelli, 2005; Li & Wang, 2009; Rehman & Wang, 2012; Wang et al., 2016), and no-reference (NR) methods (Moorthy & Bovik, 2011; Mittal et al., 2012; Kang et al., 2014; Yang et al., 2022; Zhang et al., 2023). However, a scalar score alone merely rating the overall quality without capturing regional differences or providing further information about the underlying perceptual quality, which has motivated the emergence of explainable IQA methods that aim to identify distortion types and regions while providing explanations related to the perceptual quality. Q-Bench (Wu et al., 2023) first explores the explainable IQA problem and provides a standardized benchmark for assessing explanation quality, facilitating fair comparisons across models. Based on Q-bench, Q-Instruct (Wu et al., 2024a) leverages instruction-tuned vision–language models to simultaneously evaluate image quality and provide distortion-specific explanations in natural language, highlighting the potential of MLLMs for explainable IQA.

To equip MLLMs with quality-aware perceptual and assessment abilities, several supervised fine-tuning (SFT) datasets for quality evaluation have been proposed (Wu et al., 2024a; Huang et al., 2024; Wu et al., 2024b; Jia et al., 2024). Furthermore, to assess the quality-related abilities of MLLMs, researchers have proposed several dedicated benchmarks (Wu et al., 2023; Huang et al., 2024; Zhang et al., 2025a;b;c). Q-Instruct (Wu et al., 2024a) provides large-scale instruction–response pairs targeting low-level visual perception, such as blur, noise, and distortions, to improve the perceptual abilities of multi-modal foundation models. AesExpert (Huang et al., 2024) focuses on image aesthetics perception by aligning images with human aesthetic ratings and descriptions, thereby enabling models to better capture aesthetic preferences and produce human-aligned quality assessments. In this paper, we focus on these two datasets and explore the data quality and data selection problem for explainable IQA.

## 2.2 Data Selection for Instruction Tuning

Data selection has become an increasingly hot topic in the training of large-scale models, as not all samples contribute equally to model performance. In the domain of large language models (LLMs), previous works focus on utilizing pre-defined rules (Cao et al., 2023) or gradient-based calculation (Ankner et al., 2024) to select high-value data. Inspired by these advances, recent research in vision-language models (VLMs) has placed growing emphasis on how to curate multimodal data for more effective alignment. One representative direction shows that the model itself can act as a strong filter (Chen et al., 2024), automatically screening out noisy or low-quality data to enhance instruction tuning. Another line of work considers concept-skill transferability (Lee et al., 2024), aiming to select training samples that encourage generalization across a broad range of visual-linguistic capabilities. ICONS (Wu et al., 2024c) introduces an influence-consensus mechanism that integrates multiple estimators to more reliably identify impactful samples. Collectively, these studies indicate that data selection has evolved from simple filtering to more principled and systematic strategies. However, all these studies focus on general-purpose visual question answering, while quality-related aspects remain underexplored. To address this gap, we investigate the problem of SFT data selection in the context of quality assessment.

## 3 Clustering-based Data Selection Pipeline

As experiments have demonstrated the redundancy of the current IQA instruction tuning dataset Q-Instruct, hence an efficient and powerful data selection framework is significantly needed for explainable image quality assessment. Following the common cluster-based data selection pipeline pp in data selection area, we conclude and transform the data selection framework into three stages: (1) Clustering Feature Extraction, (2) Cluster Quotas Allocation, and (3) Intra-cluster Sampling. Based on this framework, we comprehensively explore the possible strategies within our framework. Concretely, we divide the cluster features into model-related features and model-independent features and evaluate 9 combinations of features. For cluster quota allocation, we evaluate 11 allocation strategies derived from 3 key cluster metrics. For the cluster sampling method, we comprehensively

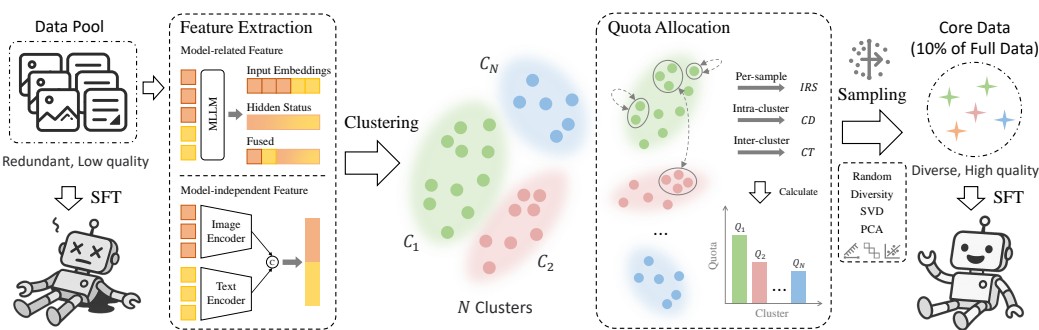

Figure 1: Overview of our proposed **IQA-Select** framework for automatically selecting high-value explainable image quality assessment question-answer samples through the efficient clustering-sampling pipeline.

evaluate the 3 sampling methods including Greedy MMD, SVD and PCA. In the rest of this section, we will introduce these stages in detail one-by-one.

### 3.1 PROBLEM DEFINITION

Given a pre-train MLLM model $M$ with parameters $\theta$ and an explainable image quality assessment dataset $Q = [z_1, z_2, ..., z_N]$ in instruction-following format, where each sample $z_i = (q_i, a_i)$ contains an input question $q$ and the corresponding answer $a$. Our problem is to find the best small instruction tuning dataset, a subset of $Q$, which can obtain the best performance for the explainable visual quality assessment task.

### 3.2 DATA CLUSTERING FEATURE SELECTION

To select diverse and informative samples from large datasets, choosing an appropriate feature representation for clustering is very important, hence we comprehensively explore the possible choices of clustering features for explainable IQA.

We first divide the clustering features into two classes: (I) model-related features and (II) model-independent features. Model-related features are based on the assumption that we already know which model we are going to fine-tune, then we can extract features specifically from the model itself. Model-independent features are more robust because we do not need to know the model and we can directly select a coreset subset using model-independent features, however the performance is usually low than the model-related features.

The model-related features contain: (1) Last Pooling, which denotes the pooling features of the last output layer of the MLLM model. (2) Last Token, which denotes the last token from the last output layer of the MLLM model. (3) Last Pooling & Vision Text Emb, which denotes the concatenated features of both the pooling features of the last output layer of the MLLM model and the visual-textual features input to the language model in the MLLM. (4) Last Token & Vision Text Emb., which denotes the concatenated features of the last token from the last output layer of the MLLM model and the visual-textual features input to the language model in the MLLM. (5) LMM feat., which denotes the three-layer features extracted from the shallow to the deep layers in the MLLM. (6) LMM Feat. & Vision Text Emb., which denotes the concatenated features of the three-layer features extracted from the shallow to the deep layers and the visual-textual features input to the language model in the MLLM. The model-independent features contain: (7) IQA Feat., which denotes the quality-related features extracted from popular IQA model (Yang et al., 2022). (8) Dino & E5 Feat., which denotes the extracted Dino-V3 vision features (Siméoni et al., 2025) of all images and the E5 text features (Wang et al., 2024) of all the question-answer pairs. (9) Dino & E5 & IQA Feat., which denotes the concatenated features of the IQA features, the Dino-V3 features, and the E5 text features.

## 3.3 Cluster Quotas Allocation Strategy Selection

Based on the calculated cluster features, we select the three-layer features extracted from the shallow to the deep in the MLLM & the visual-textual features input to the language model in the MLLM as our final cluster features. Then we explore more cluster quotas allocation strategy instead of evenly selecting sample according to the cluster scales. We first introduce three key characteristics of describing each cluster:

(1) Density (Den): The density $D$ calculates the average gaussian kernel distance of all samples inside a cluster, showing the diversity of the cluster. Concretely, $D$ is computed by:

$$D_i = \frac{1}{|C_i|(|C_i| - 1)} \sum_{m,n \in C_i, m \neq n} d(m, n),$$ (1)

where $m$ and $n$ denote two different data sample inside a cluster $C_i$, $d(m, n)$ denotes the gaussian kernel distance between sample $m$ and $n$. To ensure the diversity of our selected data, we allocate more samples for the clusters with lower density in our experiments.

(2) Instruction relevance score (IRS): The IRS (Safaei et al., 2025) evaluates how much the question $Q$ contributes to synthesizing the ground-truth answer $A$. Concretely, assuming an instruction tuning sample is denoted as a triplet $(I, Q, A)$, where $I$ denotes the input image for quality assessment, $Q$ and $A$ denote the question and answer, respectively. The IRS is computed by comparing the pre-trained MLLM's next-token cross-entropy (CE) loss with and without the question $Q$ as part of the input, which is shown as follows:

$$\text{IRS} = \frac{\mathcal{L}_{A|Q,I}}{\mathcal{L}_{A|I}}, \quad \mathcal{L}_{A|Q,I} = -\frac{1}{|\mathbf{t}^A|} \sum_{j=1}^{|\mathbf{t}^A|} \log P_\theta \left( t_j^A \mid I, Q, \mathbf{t}_{<j}^A \right),$$

$$\mathcal{L}_{A|I} = -\frac{1}{|\mathbf{t}^A|} \sum_{j=1}^{|\mathbf{t}^A|} \log P_\theta \left( t_j^A \mid I, \mathbf{t}_{<j}^A \right),$$ (2)

where $t^A$ denotes the tokenized tokens for $A$, $P_\theta$ denotes the predicted probability distribution of the pre-trained MLLM. Following this definition, a higher IRS indicates that the MLLM is more difficult to answer correctly. In our experiments, we allocate more samples for the clusters with averagely higher IRS.

(3) Transferability (Trans): Transferability measures how well the knowledge learned in this cluster can be transferred to other clusters. Following (Chen et al., 2023), we can utilize the distances between clusters to compute transferability:

$$T_i = \frac{\sum_{j=1}^N M_{ij} S_{ij}}{\sum_{j=1}^N M_{ij}}, \quad S_{ij} = \frac{x_i^\top x_j}{\|x_i\| \|x_j\|}, i, j = 1, \ldots, N,$$

$$M_{ij} = \begin{cases} 1, & S_{ij} \leq \tau, \\ 0, & S_{ij} > \tau, \end{cases}$$ (3)

where $x_i$ and $x_j$ denote the centroid embedding of cluster $i$ and $j$, $S_{ij}$ denote the cosine similarity of $x_i$ and $x_j$, $M_{ij}$ is a filtering function. In our experiments, we allocate more samples to the clusters with higher transferability.

(4) Text transferability (Text Trans): Similarly to transferability, the text transferability utilizes only the text embeddings from the MLLM for calculation. In our experiments, we allocate more samples to the clusters with higher text transferability.

Based on the four features, we use these features individually or in combination to set quotas for each cluster, and then calculate the performance after fine-tuning. Concretely, we select 11 different combinations: (1) Density, (2) IRS, (3) Transferability, (4) Text Transferability, (5) Density & IRS, (6) Transferability & IRS, (7) Text Transferability & IRS, (8) Transferability & Density, (9) Text Transferability & Density, (10) Transferability & Density & IRS, and (11) Text Transferability & Density & IRS to explore meaningful quota allocation strategies.

| Model | Question Types | | | Quadrants of Low-level Concerns | | | | Overall |
|---|---|---|---|---|---|---|---|---|
| | Yes-or-No↑ | What↑ | How↑ | Distortion↑ | Other↑ | In-context Distortion↑ | In-context Other↑ | ↑ |
| Random Guess | 50.00 | 27.86 | 33.31 | 37.89 | 38.48 | 38.28 | 35.82 | 37.80 |
| AesExpert (Huang et al., 2024) | 73.27 | 64.38 | 53.75 | 70.03 | 73.38 | 73.68 | 77.96 | 64.15 |
| Q-Instruct (Wu et al., 2024a) | 76.18 | 66.37 | 57.61 | 65.18 | 67.59 | 73.06 | 71.53 | 67.09 |
| PhotoEye (Qi et al., 2025) | 80.01 | 76.10 | 67.02 | 74.32 | 74.59 | 77.30 | 81.22 | 74.50 |
| Pretrained Model | 80.34 | 80.68 | 67.79 | 71.67 | 78.59 | 69.90 | 84.91 | 75.59 |
| Full Finetuning | 84.21 | 85.43 | 65.66 | 76.69 | 75.22 | 78.72 | 83.11 | 77.73 |
| 80% SFT data | 83.04 | 86.43 | 62.86 | 76.77 | 75.05 | 76.91 | 81.03 | 76.92 |
| 50% SFT data | 84.87 | 85.29 | 66.37 | 77.08 | 75.85 | 77.82 | 84.62 | 78.19 |
| 30% SFT data | 85.29 | 83.26 | 68.09 | 75.33 | 77.64 | 77.87 | 84.69 | 78.19 |
| 20% SFT data | 85.08 | 83.29 | 70.49 | 76.95 | 78.06 | 76.87 | 86.60 | 78.93 |
| 10% SFT data | 84.19 | 81.04 | 72.02 | 73.79 | 77.89 | 78.83 | 85.82 | 78.13 |
| 5% SFT data | 84.56 | 82.95 | 68.11 | 77.74 | 75.83 | 76.01 | 84.58 | 78.13 |
| 3% SFT data | 82.93 | 79.23 | 72.38 | 70.24 | 79.59 | 76.68 | 86.21 | 77.12 |
| 1% SFT data | 80.76 | 78.65 | 71.10 | 71.21 | 79.64 | 73.21 | 83.28 | 76.25 |

Table 1: The impact of randomly selecting instruction tuning data with different ratios for training on the performance of the large multi-modal model. Comparing to the old mllm models, current popular mllm model, i.e. InternVL3-Instruct-8B (Zhu et al., 2025) already achieves very high performance in Q-bench.

## 3.4 CLUSTER SAMPLING STRATEGY SELECTION

After having determined which features to use for clustering and how to allocate quotas for each cluster, we explore the sampling strategy for clusters. The sampling strategy is also very important because it defines how we select meaningful samples within a cluster. Concretely, we explore three distinct sampling strategies:

(1) Greedy Maximin Mean Discrepancy Sampling (Greedy MMD): Given a cluster $C_i$ and the quota $N_i$, we first calculate the squared maximum mean discrepancy between the cluster $C_i$ and the sampled data set $C_i'$, which is defined as:

$$\text{MMD}^2 = A(C_i, C_i) + A(C_i', C_i') - 2A(C_i, C_i'),$$

$$A(C_i, C_j) = \frac{1}{|C_i||C_j|} \sum_{p \in C_i, q \in C_j} d(p, q), \tag{4}$$

where $d(p, q)$ denotes the gaussian kernel distance between sample $p$ and $q$. Then greedy search is used to iteratively add samples from cluster $C_i$ to $C_i'$.

(2) Singular Value Decomposition-Based Sampling (SVD): Given a feature matrix $X_k$ which contains all the features within a cluster $C_k$ and its singular value decomposition term $X_k \approx U_k S_k V_k^T$. We calculate the leverage score $l_i$ that denotes the representativeness of sample $i$ in all subspaces of features, then we select the k samples with the highest leverage scores as the chosen samples:

$$X_k \approx U_k S_k V_k^\top, \qquad l_i = \|U_k(i, :)\|_2^2, \qquad \mathcal{S} = \underset{i}{\text{Top-K}}(l_i), \tag{5}$$

where $l_i$ is the leverage score, $S$ is the selected top-k subset.

(3) Principal Component Analysis-Based Sampling (PCA): Given a feature matrix $X_k$ that contains all features within a cluster $C_k$, principal component analysis is used to calculate the main principal subspaces. Then we obtain representatives scores $s$ by computing the projection energy on the top-k principal subspace.

$$X_c \approx U_k S_k V_k^\top, \quad Z = X_c V_k,$$

$$s_i = \|Z_i\|_2^2, \quad \mathcal{S} = \underset{i}{\text{Top-K}}(s_i), \tag{6}$$

where $s_i$ is the representatives score for sample $i$ in cluster $C_k$, $S$ is the selected top-k subset.

## 4 EXPERIMENTS

We first evaluate our proposed IQA-Select method on the Q-instruct (Wu et al., 2024a) dataset and evaluate the final performance on the Q-bench (Wu et al., 2023), which is a common MLLM benchmark designed for low-level image quality understanding. Then we conduct an in-depth analysis of

| Model | Question Types | | | Quadrants of Low-level Concerns | | | | Overall |
|---|---|---|---|---|---|---|---|---|
| | Yes-or-No↑ | What↑ | How↑ | Distortion↑ | Other↑ | In-context Distortion↑ | In-context Other↑ | ↑ |
| InternVL3-8B-Instruct | 80.34 | 80.68 | 67.79 | 71.67 | 78.59 | 69.90 | 84.91 | 75.59 |
| Full Data | 84.21 | 85.43 | 65.66 | 76.69 | 75.22 | 78.72 | 83.11 | 77.73 |
| Random 10% | 84.19 | 81.04 | 72.02 | 73.79 | 77.89 | 78.83 | 85.82 | 78.13 |
| Coincide Lee et al. (2024) 10% | 86.10 | 80.29 | 71.67 | 73.63 | 78.98 | 77.81 | 84.97 | 78.14 |
| *I. Cluster Features* | | | | | | | | |
| (1) Last Pooling | 84.30 | 81.76 | 71.62 | 74.32 | 78.04 | 78.73 | 85.82 | 78.33 |
| (2) Last Token | 85.68 | 81.71 | 70.37 | 75.43 | 77.73 | 77.49 | 86.36 | 78.46 |
| (3) Last Pooling & Vision Text Emb. | 84.24 | 81.84 | 72.72 | 75.64 | 79.11 | 77.45 | 86.21 | 78.86 |
| (4) Last Token & Vision Text Emb. | 84.12 | 81.34 | 71.23 | 74.90 | 76.82 | 78.02 | 85.84 | 77.99 |
| (5) LMM Feat. | 84.50 | 82.01 | 72.08 | 74.64 | 79.54 | 77.25 | 86.69 | 78.66 |
| (6) LMM Feat. & Vision Text Emb. | 85.59 | 82.40 | 71.28 | 75.50 | 78.45 | 79.25 | 85.82 | **78.93** |
| (7) IQA Feat. | 84.96 | 81.35 | 71.71 | 73.89 | **80.19** | 76.97 | 86.30 | 78.53 |
| (8) Dino & E5 Feat. | 84.40 | 81.84 | 71.22 | 76.13 | 78.73 | 75.91 | 85.84 | 78.60 |
| (9) Dino & E5 & IQA Feat. | 84.17 | 82.65 | 71.85 | 75.35 | 77.77 | 79.02 | 86.08 | 78.73 |
| *II. Cluster Quota Allocation* | | | | | | | | |
| (1) Density | 84.20 | 81.21 | 72.30 | 74.47 | 79.22 | 76.06 | 87.19 | 78.33 |
| (2) IRS | 85.01 | 82.10 | 72.42 | 75.99 | 79.22 | 78.35 | 85.80 | 79.13 |
| (3) Transferability | 85.31 | 81.29 | 70.17 | 74.15 | 78.23 | 77.44 | 85.88 | 78.13 |
| (4) Text Transferability | 84.01 | 81.84 | 70.93 | 73.23 | 77.63 | 77.53 | 84.97 | 77.79 |
| (5) Density & IRS | 84.64 | **82.48** | 72.30 | **76.13** | 78.08 | 78.35 | 86.67 | 79.00 |
| (6) Transferability & IRS | 84.41 | 82.18 | 71.33 | 76.11 | 77.31 | 76.97 | 85.90 | 78.33 |
| (7) Text Transferability & IRS | 83.56 | 80.51 | 71.33 | 74.66 | 77.74 | 76.40 | 85.06 | 77.73 |
| (8) Transferability & Density | 85.75 | 82.07 | 71.88 | 76.03 | 79.06 | 77.82 | 86.69 | **79.20** |
| (9) Text Transferability & Density | 85.54 | 82.20 | 71.71 | 75.94 | 77.45 | 78.73 | 87.14 | 78.93 |
| (10) Transferability & Density & IRS | 84.02 | 81.43 | 71.94 | 75.40 | 78.65 | 77.11 | 85.34 | 78.46 |
| (11) Text Transferability & Density & IRS | 84.87 | 81.43 | 71.19 | 75.96 | 78.61 | 76.25 | 85.82 | 78.60 |
| *III. Cluster Sampling* | | | | | | | | |
| (1) Greedy MMD sampling | 85.11 | 80.68 | **73.73** | 74.03 | 78.71 | **79.64** | 86.97 | 78.93 |
| (2) SVD sampling | 85.91 | 82.12 | 72.77 | 75.71 | 79.17 | 77.82 | 88.38 | **79.40** |
| (3) PCA sampling | 84.41 | 81.90 | 72.20 | 75.48 | 77.36 | 77.63 | 87.54 | 78.53 |
| *Final Results* | | | | | | | | |
| I-(6) + II-(8) + III-(2) | **85.91** | 82.12 | 72.77 | 75.71 | 79.17 | 77.82 | **88.38** | **79.40** |

Table 2: The impact of selecting different clustering features, cluster quota allocation strategies and cluster sampling strategies for data selection on the performance of the large multi-modal model. 10% data is selected from the original Q-Instruct instruction tuning dataset. The best and runner-up performances are bold and underlined, respectively.

our method and experimental results. Additionally, we evaluate the robustness of our IQA-Select on the Aesexpert (Huang et al., 2024) dataset and report the performance in Aesbench.

## 4.1 EXPERIMENTAL SETUP

**Dataset and Evaluation Metric.** For original instruction tuning dataset, we select Q-instruct and Aesexpert dataset as the testbeds of our IQA-Select method. The Q-instruct dataset consists of about 200K instruction tuning examples, covering quality reasoning data and low-level visual quality answering data across various distortion types. The Aesexpert dataset contains 409K instruction tuning examples covering various aesthetic problems such as composition, color, lighting, and clarity. For evaluation, we strictly follow the open-source evaluation tool VLMEvalKit and report the model performance on the public part of the Q-bench and AesBench.

**Implementation Details.** In our experiments, we select a cutting-edge MLLM, InternVL3-8B-Instruct, as a baseline model for training. The InternVL3 model consists of a vision encoder, a feature projector, and a large language model. Low rank adaptation (LoRA) is adopted during the training and the LoRA rank parameter is set to 16. The whole training is conducted for 1 epoch. The learning rate is set to $2 \times 10^{-5}$ with a cosine decay schedule. All the experiments are conducted on a single H200 GPU.

## 4.2 EXPERIMENTAL RESULTS FOR IQA DATA SELECTION

### 4.2.1 DO WE NEED ALL INSTRUCTION TUNING DATA FOR EXPLAINABLE IQA?

In this section, we discuss the necessity of utilizing all the instruction-tuning dataset for explainable IQA. To cope with this problem, we first select the recently popular state-of-the-art MLLM, InternVL3-Instruct, and evaluate its generalization ability on the Q-Bench. The result is shown in Table 1, we can observe that as the development of MLLMs, the pre-trained InternVL3-Instruct

| Method | AESA ↑ | AESE ↑ | AESP ↑ | Overall ↑ |
|---|---|---|---|---|
| Baseline Model | 27.5 | 62.5 | 76.67 | 55.56 |
| Full Data | **33.33** | 63.33 | 80.00 | 58.89 |
| Random 10% | 30.83 | 65.00 | 82.50 | 59.44 |
| Ours 10% | 31.67 | **66.67** | **85.00** | **61.11** |

Table 3: Performance of coreset selection on the AesBench VAL benchmark. We fine-tune the InternVL3-Instruct using coresets with a 10% sampling ratio. The best and runner-up performances are bold and underlined, respectively.

can already achieve very high performance (overall 75.59%) compared to the specifically fine-tuned model Q-Instruct (overall 67.09%).

Then from Table 1, we can observe that utilizing the 100% supervised fine-tuning (sft) data for training can only achieve about 2% improvements (overall 77.73% vs 75.59%), while the randomly selected 20% sft data achieve higher performance. The reason is because the base performance of InternVL3 is relatively high, making the space of improvement limited, and the Q-Instruct dataset is redundant, fine-tuning the MLLM with all dataset may inversely cause the MLLM to forget the knowledge it learned before. From the results of Table 1, as the ratio of randomly selected data increases, the performance curve of the fine-tuned model is approximately an inverted U-shaped curve, demonstrating that the scale of the instruction tuning data should neither be too large nor too small.

### 4.2.2 IMPACT OF DIFFERENT CLUSTERING FEATURES

The experimental results are summarized in Table 2. First, we can observe that the model-related features averagely achieve higher performance than model-independent features, which means that it is better to select a model-specific feature for optimizing the specific model. Second, we can observe that the LMM Feat. provides meaningful features for clustering and combining the LMM Feat. and Vision Text Emb. achieves the best performance (oervall 78.93%) on Q-Bench, demonstrating that utilizing more layers of features leads to a better performance.

### 4.2.3 IMPACT OF CLUSTER QUOTAS ALLOCATION

We also report the impact of cluster quota allocation in Table 2. First, we can observe that the Transferability & Density combination achieves the best performance in Q-Bench. It is mainly because merely utilizing the transferability may allocate more quotas to representative clusters with poor inner diversity, hence with the help of density, it can reasonably allocate the quota for each cluster. Second, we find that the IRS feature itself is very meaningful; however, combining IRS with other features leads to poorer performance. Third, we can observe that the performance of using transferability is consistently better than text transferability, demonstrating that combining the vision and text features from the MLLM can better compute the generalization ability of each cluster to other clusters.

### 4.2.4 IMPACT OF DIFFERENT CLUSTER SAMPLING STRATEGIES

According to the summarized results in Table 2, we can find that the SVD sampling performs the best comparing to greedy mmd sampling and PCA sampling. Among the three sampling strategies, greedy mmd sampling tends to sample diverse data inside the feature space, while SVD and PCA sampling tend to find the representative data in each cluster. It can be observed that selecting representative data is more important than selecting diverse data in the explainable image quality area. This is because representative data can help the MLLM to learn the typical data sample and prevents the selection of noisy outliers or mislabeled data. We can also find that SVD sampling can select the representative data more effectively than PCA sampling.

### 4.2.5 OVERALL DISCUSSION OF IQA-SELECT

**Performance on explainable image quality assessment.** After comprehensively exploring the three stages of clustering-based explainable IQA data selection pipeline, our final method uses LMM

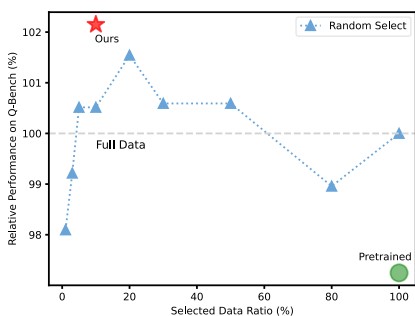 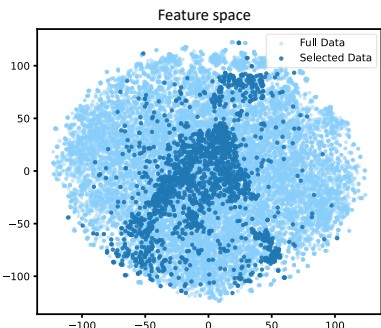

Figure 2: Left: The performance of our **IQA-Select** compared with randomly selecting baselines and pre-trained MLLM baseline. Right: the visualization of selected data and full data in feature space.

features and Vision Text embeddings for clustering, using the combination of transferability and density for cluster quota allocation, and SVD sampling for cluster sampling. The final method achieves considerably higher performance (overall 79.40%) than randomly selecting 10% instruction tuning data (overall 78.13%).

**Performance on explainable image aesthetic assessment.** In addition, we also evaluate the effectiveness of our IQA-Select method in the AesBench and select 10% instruction tuning data from the AesExpert dataset. The results are summarized in Table 3. First, we can observe that the data redundancy problem also happened in the explainable image aesthetic assessment area, randomly selecting 10% data for fine-tuning achieves better performance than using 100% instruction tuning data (overall 59.44% vs 58.89%). Second, our IQA-Select method performs better than randomly selecting 10% data (overall 61.11% vs 59.44%), demonstrating the generalization and effectiveness of our IQA-Select framework.

**Visualization of the selected samples in the feature space.** To demonstrate the effectiveness of our IQA-Select method, we use T-SNE to visualize the selected samples of IQA-Select and the full samples in the MLLM's feature space. The result is visualized in Figure 2, we can see that our IQA-Select method focuses mainly on selecting the representative features which are in the center of the feature space, but it can also sample several unique samples at the edge of the circle to ensure the diversity of final sample set.

**Limitations.** Our paper also has its limitation, the performance of current MLLM on Q-Bench is already high, making the improvement of full fine-tuning limited (overall 77.73% vs 75.59%). Future work includes extending the data quality assessment problem to more difficult explainable quality assessment area, such as explainable video quality assessment.

## 5 CONCLUSION

In this paper, we systematically investigate the role of data quality for explainable IQA. Using a powerful pre-trained MLLM, we first investigate the changes in model performance after fine-tuning with different sizes of instruction tuning data. We find that selecting a subset of the data set randomly using an appropriate ratio can even lead to better results than training with the entire instruction tuning dataset. Beyond randomly sampling a subset, we propose a clustering-based data selection framework with three stages: clustering feature extraction, cluster quota allocation, and cluster sampling strategy. Then we systematically analyze the choices of each stage and propose a simple but efficient data selection method IQA-Select for explainable IQA. The experimental results demonstrate that IQA-Select can achieve 102.1% and 103.7% performance of full fine-tuning using only 10% selected data in Q-Bench and AesBench respectively, significantly reducing computational costs while achieving better performance. We hope that our paper can provide a new perspective for future research on exploring the quality of instruction tuning data for explainable IQA.

# ETHICS STATEMENT

Our work adheres to the ICLR Code of Ethics. This work explore the data quality and data selection for explainable image quality assessment. All data used in this paper are publicly available datasets, which contains no sensitive, personal content.

# REPRODUCIBILITY STATEMENT

We have taken considerable measures to ensure the reproducibility of our work. The main paper discusses challenges of data quality and quantity in explainable image quality assessment, introduces a data selection pipeline, and reports exploratory experiments on two explainable image quality assessment datasets. We plan to release the final data selection method and the selected SFT data from two datasets upon publication to facilitate replication and follow-up research.

# LLM USAGE STATEMENT

Large language models (LLMs) were only used for information retrieval and minor sentence polishing, and were not involved in writing the paper.

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

## A APPENDIX

### A.1 QUALITATIVE RESULTS

To demonstrate the effectiveness of our IQA-Select framework, we visualize the model outputs before and after the finetuning process in Figure 3. From the results, we can observe that the large multi-modal model is able to identify more distortion and image composition problems, showing the limitations of current general large multi-modal models.

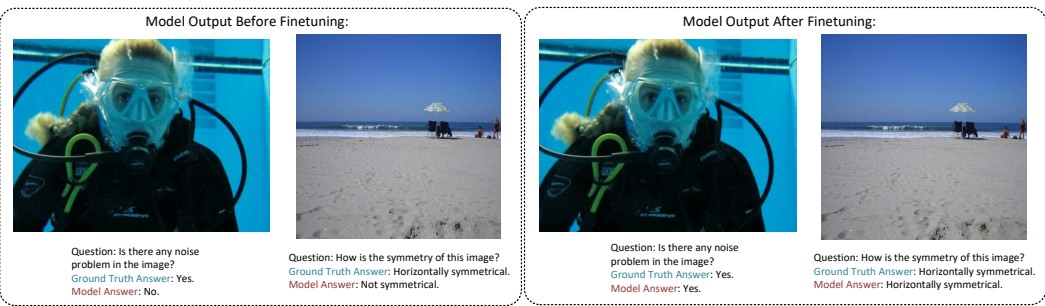

Figure 3: Visualization of the model predictions before and after finetuning on Q-Bench. After finetuning with our instruction data selected by our IQA-Select framework, the InternVL3-8B model is able to perceive more quality problems.

### A.2 VISUALIZATION OF SELECTED DATA

To validate the effectiveness and redundancy of the current explainable image quality assessment instruction tuning data Q-Instruct, we concretely visualize the selected samples and filtered samples by our IQA-Select method. The whole examples are shown in Figure 4. From the figure, we can observe that IQA-Select can effectively filter out similar question-answer pairs (i.e. image sharpness) and easy question-answer pairs (i.e. determining whether the image is distorted). Meanwhile, IQA-Select can precisely select the unique samples in the Q-Instruct dataset. For example, IQA-Select localizes that the number of image with good quality is limited and can precisely select it.

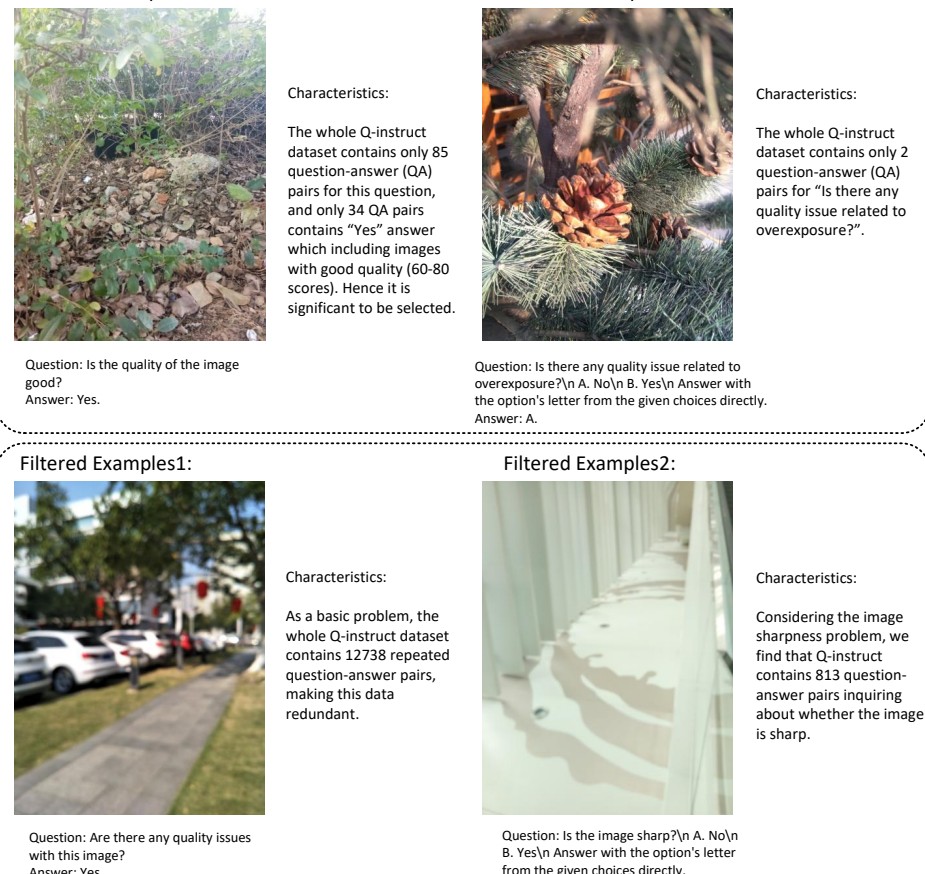

Figure 4: Visualization of the filtered samples and selected samples of our IQA-Select framework. We concretely display the characteristics for each question-answer sample.

