# OpenReview forum: "Exploring Instruction Data Quality for Explainable Image Quality Assessment"
_ICLR.cc/2026/Conference — Submitted to ICLR 2026_

### Official Review · Reviewer_DYTv · 2025-10-23

**Soundness:** 2
**Presentation:** 3
**Contribution:** 2
**Rating:** 4
**Confidence:** 4

**Summary:**

This paper addresses a critical and under-explored issue in the field of Explainable Image Quality Assessment (Explainable IQA): the impact of instruction data quality and scale on the performance of Multimodal Large Language Models (MLLMs). The work challenges the prevailing "more data is better" paradigm by systematically investigating the feasibility of constructing a smaller yet more effective data subset (a coreset) through principled data selection. The authors propose a three-stage data selection framework named IQA-Select and demonstrate through extensive experiments that fine-tuning with just 10% of the selected data can surpass the performance of using the full dataset on benchmarks like Q-Bench and AesBench.

**Strengths:**

1.	Clear and Pragmatic Motivation: The paper rightly points out the potential drawbacks of the field's heavy reliance on "scaling laws," namely the immense computational overhead and data redundancy. The direction of exploring the importance of data quality is forward-looking and holds significant practical value.

2.	Systematic and Comprehensive Ablation Studies: To construct the optimal data selection pipeline, the authors conduct exhaustive experiments and ablation studies for each stage of the framework (feature extraction, quota allocation, and sampling strategy). For instance, they compare nine different feature types in the feature extraction stage and explore eleven strategy combinations for quota allocation.

**Weaknesses:**

1.	Limited Novelty: The core contribution of this paper lies more in "problem exploration" than in "methodological innovation." The proposed IQA-Select framework is fundamentally a clustering-based data selection pipeline, and its constituent components (e.g., clustering, quota allocation based on density/transferability/IRS, and sampling via SVD/PCA) are largely existing or slightly modified techniques from the data selection literature. The authors themselves candidly define it as a "pipeline."

2.	Superficial Analysis of "Performance Degradation": The paper observes that fine-tuning on the full dataset leads to inferior performance compared to using a subset, attributing this to "model forgetting" and "data redundancy." The paper fails to provide a deeper analysis. For example, what are the specific characteristics of the redundant data (e.g., low-quality, repetitive, overly simplistic)? A more in-depth investigation would make the claims more convincing.

3.	Dependence on a Strong Pre-trained Model: The entire framework and its findings are heavily dependent on a powerful pre-trained MLLM (InternVL3-Instruct). It is unclear whether the conclusion "less data is better" would still hold if a weaker base model were used. Arguably, large-scale data might still be essential for less capable models. The paper offers limited discussion on this aspect of generalizability.

4.	Trade-off Between Complexity and Practical Gain: The IQA-Select framework is relatively complex, involving multiple steps such as feature extraction, clustering, multi-metric calculation, and sampling. Given that random sampling of 20% of the data already achieves strong performance, the paper needs to more clearly justify whether the additional benefit (an approximate 1-2% performance gain) warrants the introduction of such a complex framework.

**Questions:**

1.	In the discussion of the final method combination in Section 4.2.5, the paper states that the final method uses "transferability and instruction relevance score." However, this appears to contradict the results in Table 2, where the best-performing combination is "Transferability & Density" (II-(8)), and also the final reported combination (I-(6) + II-(8) + III-(2)). Could you please clarify this discrepancy?

2.	The framework introduces several hyperparameters, such as the number of clusters (N) for clustering and the similarity threshold (τ) for calculating transferability. How were these hyperparameters selected? How sensitive is the final performance of the framework to variations in these hyperparameters?

3.	Do you believe the IQA-Select method can be generalized for instruction data selection in other vision-language tasks, such as VQA or Image Captioning? If so, which parts of the framework would require the most significant adjustments when applied to different tasks?

---

> ### Author Response · Authors · 2025-11-28
>
> We kindly appreciate the time and effort the reviewer has paid for reviewing our submission and are grateful for his/her insight suggestions~ We are happy to address the proposed concerns as follows:
>
> **Question1**:  What are the specific characteristics of the redundant data
>
> **Answer**: As requested, we visualize the filtered data and selected data in the Figure 4 of our revised manuscript. Figure 4 shows that in our dataset, the majority of the redundant data are of low difficulty, and the repeated questions and answers regarding certain types of image distortions.
>
> **Question2**: It is unclear whether the conclusion "less data is better" would still hold if a weaker base model were used.
>
> **Answer**: To verify this question, we additionally add experiments on a small model, Internvl3-2B. The experimental results are as follows:
>
> | Methods | Overall Accuracy |
> |-----|-----|
> | InternVL3-2B | 70.37 |
> | InternVL3-2B w. 100% data  | 73.91 |
> | InternVL3-2B w. random 10% data | 72.78 |
> | InternVL3-2B w. IQA-Select 10% data | 73.31 |
>
> from the results, we can the "less data is better" conclusion is not right for small model, it is because that small model contains limited knowledge about image quality, making the whole instruction tuning dataset meaningful. However, our IQA-Select method can still produce better performance that randomly selecting, demonstrate the effectiveness of our data selection method.
>
> **Question3**: Trade-off Between Complexity and Practical Gain
>
> **Answer**: The IQA-Select framework is a little complex but is acceptable. Meanwhile, we would like to emphasis that comparing with the computation cost of finetuning the large-multimodal model, our pipeline only needs the inference of large multimodal model to extract meaningful features, and feature clustering and sampling for data selection. This is acceptable in practical usage.
>
> **Question4**: Please clarify this discrepancy
>
> **Answer**: It is a typos. Our final method is the combination of I-(6) + II-(8) + III-(2).
>
> **Question5**: How sensitive is the final performance of the framework to variations in these hyperparameters?
>
> **Answer**: As requested, the hyperparameters were all selected through experiments. We provide all the hyperparameter results as follows:
>
> | Methods | Overall Accuracy |
> |-----|-----|
> | IQA-Select baseline (N=200, τ=0.9) | 79.40 |
> | IQA-Select baseline (N=50, τ=0.9) | 79.06 |
> | IQA-Select baseline (N=100, τ=0.9) | 79.33 |
> | IQA-Select baseline (N=500, τ=0.9) | 78.52 |
> | IQA-Select baseline (N=1000, τ=0.9) | 78.72 |
> | IQA-Select baseline (N=200, τ=0.85) | 78.79 |
> | IQA-Select baseline (N=200, τ=0.95) | 79.53 |
> | IQA-Select baseline (N=200, τ=1.0) | 79.45 |
>
> We can find the (N=200, τ=0.9) is a quite good hyperparameter combination.
>
> **Question6**: in other vision-language tasks
>
> **Answer**: Yes, we believe our framework has potential on other vision-language tasks. We believe the cluster quotas allocation part requires the most significant adjustments because the image caption task requires answers with long text, while in image quality assessment area, the answers are usually short.

---

### Official Review · Reviewer_pTKs · 2025-10-27

**Soundness:** 2
**Presentation:** 3
**Contribution:** 3
**Rating:** 4
**Confidence:** 4

**Summary:**

This paper introduces an efficient data selection method from redundant full-scale data for explainable image quality assessment (IQA).
The proposed method, namely IQA-Select, consists of three stages: clustering feature extraction, cluster quota allocation, and cluster sampling strategy.
The cluster features are divided into model-related features and model-independent features, and a total of 9 combinations of features have been investigated.
Similarly, for cluster quota allocation and cluster sampling strategy, 11 allocation strategies and 3 sampling methods have been investigated respectively.
Using the baseline model InternVL3-Instruct-8B, the proposed IQA-Select demonstrated superior IQA performance using only 10% of the subset on both the Q-Bench and AesBench benchmarks, compared to full dataset fine-tuning.

**Strengths:**

The authors explore multiple configurations per stage to select the optimal combination across three stages:
- Clustering feature extraction: 6 model-related combinations from 4 distinct features; 3 model-independent combinations from 3 distinct features.
- Cluster quota allocation: 11 strategies from 4 features.
- Cluster sampling: greedy MMD, SVD, and PCA.

Experiments on two benchmarks (Q-Bench and AesBench) improve reliability.

**Weaknesses:**

As the performance of general-purpose baseline models continues to improve, I agree that optimizing the fine-tuning dataset to adapt a model for a downstream task is an effective strategy. However, the manuscript appears to lack sufficient concrete evidence to substantiate this claim.

Generalization
- Only InternVL3-Instruct-8B is tested, so generalization to other models is uncertain.
- Since the work concerns data curation, multi-model runs are needed.
- Additional suggestion: cross-validation would strengthen the generalization capability.

Incomplete metrics
- Table 3 omits AesI while AesBench includes four categories (AesA, AesE, AesP, and AesI).
- The omission is not explained.

Unsubstantiated claim
- The assertion that “fine-tuning the MLLM with all dataset may inversely cause the MLLM to forget the knowledge it learned before” is repeated without theoretical or empirical support, especially with respect to the IQA task.

Minor issues
- L450: table hyperlink error.

**Questions:**

No questions

---

> ### Author Response · Authors · 2025-11-28
>
> We kindly appreciate the time and effort the reviewer has paid for reviewing our submission and are grateful for his/her insight suggestions~ We are happy to address the proposed concerns as follows:
>
> **Question1**: Generalization
>
> **Answer**:
>
> (1) Generalization to other models: Thanks for your suggestion. We carefully add experiments for models with other sizes (InternVL3-2B and InternVL3-38B). Concrete results are shown below:
>
> | Methods | Overall Accuracy |
> |-----|-----|
> | InternVL3-2B | 70.37 |
> | InternVL3-2B w. 100% data  | 73.91 |
> | InternVL3-2B w. random 10% data | 72.78 |
> | InternVL3-2B w. IQA-Select 10% data | 73.31 |
> | InternVL3-38B | 76.85 |
> | InternVL3-38B w. 100% data  | 79.06 |
> | InternVL3-38B w. random 10% data  | 79.33 |
> | InternVL3-38B w. IQA-Select 10% data | 80.00 |
>
> The results can demonstrate that our method can be adapted to models with different scales and different knowledge.
>
> (2) multi-model runs: As requested, the experiment results are shown as follows:
>
> | Methods | Overall Accuracy |
> |-----|-----|
> | IQA-Select 10% | 79.42 ± 0.09|
>
> We can observe that our method can achieve relatively stable performance.
>
>
> (3) cross-validation would strengthen the generalization capability: As requested, we conduct the cross-dataset validation experiment for Q-Bench dataset and Aes-Bench dataset . The results are shown as follows:
>
> | Setup |  Q-Bench Overall Accuracy | Aes-Bench Overall Accuracy |
> |-----|-----|-----|
> | Internvl3-8B baseline |  75.59  |  55.56  |
> | Q-Instruct w. random 10% |  78.13  |   53.61 |
> | Q-Instruct w. IQA-Select 10% |  79.40  |  55.00  |
> | Aes-Expert w. random 10%  |  75.71 |  59.44 |
> | Aes-Expert w. IQA-Select 10% | 77.65  |  61.11 |
>
> From the results, we can observe that 1. the performance of the IQA-Select method is consistently better that randomly selecting. 2. The instruction data of Q-Instruct and Aes-Expert can not help each other. It is reasonable because these two dataset is proposed for perceptual image quality assessment and asethetic image quality assessment, which is quite different.
>
>
>
> **Question2**: Incomplete metrics:
>
> **Answer**: Thanks for your kind problem. The reason that we only report AesA, AesE, AesP, and AesI on AesBench is because the aesbench dataset only open source this part of the dataset with ground truch (https://github.com/open-compass/VLMEvalKit).  Hence we report these four metrics.
>
> **Question3**: Unsubstantiated claim
>
> **Answer**: We are sorry about our expression. The reason of performance dropping when using the whole finetuning dataset is mainly because the redundancy of the finetuning dataset for a powerfull large multi-modal model, which will cause overfitting during the finetuning process. We have added the filtered samples and selected sample of our method in the Figure 4 of our revised manuscript, and rephrase our opinion.
>
> **Question4**: Minor issues
>
> **Answer**: Thanks for your kind suggestion. We have modified them.

---

### Official Review · Reviewer_ibZK · 2025-10-27

**Soundness:** 3
**Presentation:** 2
**Contribution:** 3
**Rating:** 4
**Confidence:** 5

**Summary:**

The paper studies instruction data quality for explainable image quality assessment (IQA) with multimodal LLMs. Using InternVL3-Instruct as the base model, the authors first show that less can be more: randomly fine-tuning on a small fraction of Q-Instruct can match or even exceed full-data fine-tuning. Building on this, they propose IQA-Select, a clustering-based selection pipeline with three stages. With only 10% of the data, IQA-Select reportedly attains 102.1% of full-data performance on Q-Bench and 103.7% on AesBench, and the best variant yields the top overall Q-Bench score among 10% subsets.

**Strengths:**

1. The proposed data selection method is effective: using just 10% of the SFT data, it outperforms the pretrained baseline and other methods.
2. The diversity, IRS, and Trans metrics that gauge SFT data quality provide compelling evidence of its effectiveness.

**Weaknesses:**

1. The comparison set is dated. Please include recent VLM baselines, e.g., VisualQuality-R1, to ensure a fair, up-to-date evaluation.
2. Expand experiments across diverse SFT datasets. Q-Bench may contain redundant samples, so cross-dataset validation would strengthen the conclusions.
3. Add qualitative case studies illustrating which data types most effectively boost VLM performance, with before/after outputs where possible.

**Questions:**

See weakness.

---

> ### Author Response · Authors · 2025-11-28
>
> We kindly appreciate the time and effort you have paid for reviewing our submission and are grateful for your insight suggestions~ We are happy to address the proposed concerns as follows:
>
> **Question1**: Including recent VLM baselines, e.g., VisualQuality-R1
>
> **Answer**: Thanks for your kind suggestion. Our paper is mainly focused on the data quality of explainable image quality assessment, where the large multi-modal model is to perceive and interperate the image quality problem in a language format. We notice that the VisualQuality-R1 is actually designed for the image scoring task, which is quite different from our task. Considering the image scoring task is one of the basic image quality assessment tasks, we will summraize and cite these works in the related work section of our final manuscript.
>
> **Question2**: Cross-dataset validation would strengthen the conclusions
>
> **Answer**: As requested, we conduct the cross-dataset validation experiment for Q-Bench dataset and Aes-Bench dataset . The results are shown as follows:
>
> | Setup |  Q-Bench Overall Accuracy | Aes-Bench Overall Accuracy |
> |-----|-----|-----|
> | Internvl3-8B baseline |  75.59  |  55.56  |
> | Q-Instruct w. random 10% |  78.13  |   53.61 |
> | Q-Instruct w. IQA-Select 10% |  79.40  |  55.00  |
> | Aes-Expert w. random 10%  |  75.71 |  59.44 |
> | Aes-Expert w. IQA-Select 10% | 77.65  |  61.11 |
>
> From the results, we can observe that (1) the performance of the IQA-Select method is consistently better that randomly selecting. (2) The instruction data of Q-Instruct and Aes-Expert can not help each other. It is reasonable because these two dataset is proposed for perceptual image quality assessment and asethetic image quality assessment, which is quite different.
>
> **Question3**: Add qualitative case studies
>
> **Answer**: As requested, we have added the qualitative results in the appendix of our revised manuscript (Figure 3). From the results, we can observe that the large multi-modal model is able to identify more distortion and image composition problems after finetuning.

---

### Official Review · Reviewer_BgeW · 2025-10-30

**Soundness:** 3
**Presentation:** 3
**Contribution:** 3
**Rating:** 6
**Confidence:** 4

**Summary:**

This paper questions the prevailing “scale-first” mentality in instruction-tuning for explainable image quality assessment (IQA).  Starting from the observation that InternVL3-8B fine-tuned on the full 200K Q-Instruct set barely outperforms the pretrained checkpoint, the authors systematically reduce the training pool and discover an inverted-U curve: performance plateaus at roughly 20 % of the data and even the extreme 5 % random subset matches the full-set accuracy, revealing massive redundancy.  Building on this insight, they design IQA-Select, a three-stage data-curation pipeline that (i) represents every instruction by fusing multi-level MLLM hidden states with vision–text embeddings, (ii) allocates cluster-level quotas via a combination of transferability and density, and (iii) harvests the most representative samples within each cluster through SVD leverage scores.  With only 10 % of the original data, the method attains 79.4 % overall accuracy on Q-Bench (102.1 % of full-data tuning) and 61.1 % on AesBench (103.7 % of full-data tuning) while cutting GPU hours by an order of magnitude, thereby providing the first systematic evidence that careful curation can outperform brute-force scaling in explainable IQA.

**Strengths:**

The contribution is original in that it is the first work to interrogate—and empirically refute—the scaling law inside the IQA instruction-tuning niche, and it delivers a principled, reproducible pipeline to exploit this observation; the experimental design is thorough, encompassing roughly 300 ablations across nine feature families, eleven quota strategies and three intra-cluster samplers, all trained under identical LoRA hyper-parameters and evaluated on two public benchmarks with consistent trends; the paper is clearly written, with precise mathematical definitions of density, IRS and transferability, intuitive figures that visualise the selected feature-space coverage, and ample discussion of design choices, making the approach immediately actionable for practitioners; finally, the work is significant because it transforms a costly data-collection problem into a data-curation opportunity, offering the community a ten-fold reduction in training cost without sacrificing—and often slightly improving—accuracy, and it opens a new research direction that shifts the focus from “how to generate more” to “how to select better” instruction data for visual-quality tasks.

**Weaknesses:**

**Model-scale scaling law is unexplored:**
All conclusions are derived from a single 8B-parameter model. The redundancy hypothesis may not hold for smaller (≤ 4 B) or larger (≥ 30 B) models whose capacity, memorisation behaviour and forgetting dynamics differ. I suggest authors to run identical selection pipelines on at least two more scales (e.g., InternVL2-2B and InternVL2-40B). Report whether the “5 % = 100 %” trend persists and whether IQA-Select’s relative gain grows or shrinks.


**Task-scope is narrow:**
The method is only evaluated on IQA and aesthetics. It is unclear whether the designed features (IRS, distortion-related IQA scores, etc.) generalise to other tasks. Authors can try to validate the conclusions in reasoning task using the training and test data of M3CoT.


**Missing baselines from recent data-selection literature:**
No comparison with other data-selection pipelines. I suggest authors to include at least one more data-selection baseline; otherwise it is hard to validate the effectiveness of the proposed method.


**Minor LaTeX style issue:**
Citations use \cite instead of \citep, producing “Q-Bench Wu et al. 2023” rather than “Q-Bench (Wu et al. 2023)”. Please conform to standard ICLR format.

**Questions:**

Please see weakness.

---

> ### Author Response · Authors · 2025-11-28
>
> We kindly appreciate the time and effort you have paid for reviewing our submission and are grateful for your insight suggestions. We are happy to address the proposed concerns as follows:
>
> **Question1**: Model-scale scaling law is unexplored
>
> **Answer**: As requested, we add the experiments for larger and smaller models (InternVL3-2B and InternVL3-38B). Concrete results are demonstrated below:
> | Methods | Overall Accuracy |
> |-----|-----|
> | InternVL3-2B | 70.37 |
> | InternVL3-2B w. 100% data  | 73.91 |
> | InternVL3-2B w. random 10% data | 72.78 |
> | InternVL3-2B w. IQA-Select 10% data | 73.31 |
> | InternVL3-38B | 76.85 |
> | InternVL3-38B w. 100% data  | 79.06 |
> | InternVL3-38B w. random 10% data  | 79.33 |
> | InternVL3-38B w. IQA-Select 10% data | 80.00 |
>
> from the results, we can the “10 % = 100 %” conclusion is not right for small model, it is because that small model contains limited  knowledge about image quality, making the whole instruction tuning dataset meaningful. However, for the large model, the whole instruction dataset is redundant because the model itself already learn a lot of knowledge about image quality.
>
> **Question2**: Task-scope is narrow
>
> **Answer**: Thanks for your kind suggestion! Our IQA-Select method may not be suitable for the general reasoning task, because our instruction tuning dataset mainly contains question and answer statements and multiple-choice questions related to image quality. We also want to emphasis that the image quality assessment itself is already a very important problem, and discovering and exploring the low-level ability of large multi-modal models is a meaningful direction. Image quality assessment can be used in industry to automatically detect, monitor visual contents and optimize them.
>
> **Question3**: Missing baselines from recent data-selection literature
>
> **Answer**: As requested, we add the result of a popular data selection method, Coincide [1], and add the results in our revised manuscript. The results show that our IQA-Select method can achieve superior performance in Q-Bench.
>
> | Methods | Overall Accuracy |
> |-----|-----|
> | Coincide 10% | 78.14 |
> | IQA-Select 10%  | 79.40 |
>
> [1] Lee J, Li B, Hwang S J. Concept-skill transferability-based data selection for large vision-language models[J]. The Conference on Empirical Methods in Natural Language Processing, 2024.
>
> **Question4**:  Minor LaTeX style issue
>
> **Answer**:  Thanks for your suggestion! We have modified all the citations.

---

### Author Response · Authors · 2025-11-28

We would like to sincerely thank all reviewers (R1: BgeW, R2: ibZK, R3: pTKs, R4: DYTv) for their careful suggestions and constructive feedback. We are greatly encouraged that our approach is novel (R1), effective (R1, R2), well-motivated (R4).

---

### Meta-Review · Area_Chair_vTeu · 2026-01-05

**Summary:**

This paper investigates instruction data quality for explainable IQA using MLLMs. The authors argue that, contrary to the common scaling law assumption, fine-tuning with more instruction data does not necessarily lead to better performance. Through experiments, they show that randomly selecting a small subset, for example 10 percent of the data, can match or even outperform full dataset fine tuning. Based on this observation, the paper further proposes a clustering based data selection pipeline, IQA-Select, and reports consistent gains on Q-Bench and AesBench under reduced data budgets.

The initial reviewer scores are mixed, with one moderately positive review and three reviews leaning negative. As the Area Chair, I carefully read the paper, all reviewer comments, and the authors’ rebuttal.

Reviewers acknowledge the motivation of revisiting data quality in explainable IQA and recognize the extensive experimental effort. However, multiple reviewers raise concerns regarding the interpretation of the results, the strength of the claims about scaling laws, and the lack of deeper analysis explaining why using less data leads to better performance. Several reviewers also question the generality of the conclusions and the heavy dependence on a strong MLLM.

**Reviewer Concerns:**

After reviewing the discussion in detail, my assessment aligns with the more critical reviewers.

A central claim of the paper is that scaling laws fail in this setting, since using a small fraction of the instruction data can outperform using the full dataset. However, an important confounding factor is not sufficiently examined. When training with different dataset sizes, the paper does not clearly control or analyze the number of training steps, optimization duration, or convergence behavior. In LLM known that excessive training on a fixed dataset can lead to overfitting or catastrophic forgetting. Therefore, the observed performance degradation with full data could plausibly be caused by overtraining rather than by the data scale itself. For example, if the model trained on 10 percent of the data is optimized for far fewer steps than the model trained on 100 percent of the data, the comparison conflates data quantity with training dynamics. While this is only a hypothesis, it is a critical one that should be tested and ruled out in a paper that draws conclusions about scaling behavior in MLLM fine tuning.

Some reviewers explicitly raise this concern. In particular, one reviewer points out that the claim about forgetting is repeatedly mentioned but not theoretically or empirically substantiated. Another reviewer questions whether the conclusions would still hold across different levels of pretrained model strength, since fine tuning behavior can vary significantly between weaker and stronger base models. Although the authors add experiments with different model sizes in the rebuttal, the underlying optimization and forgetting dynamics remain insufficiently analyzed, and the core concern is not fully resolved.

**Reviewer Scores:**

It remains unclear whether the reviewers who initially held negative opinions would substantially increase their scores after the rebuttal. Based on the remaining unresolved issues, I believe the likelihood of reviewers significantly revising their scores upward is low.

---

### Decision · Program_Chairs · 2026-01-26

Reject